# Effectiveness of drug interventions to prevent sudden cardiac death in patients with heart failure and reduced ejection fraction: an overview of systematic reviews

Muaamar Al-Gobari,[1] Sinaa Al-Aqeel,[2] François Gueyffier,[3] Bernard Burnand[1]

¹Institute of Social and Preventive Medicine (IUMSP), Cochrane Switzerland, Lausanne University Hospital (CHUV), Lausanne, Switzerland
²Clinical Pharmacy Department, College of Pharmacy, King Saud University, Riyadh, Saudi Arabia
³Laboratoire de Biologie et Biométrie Evolutive-Equipe Modélisation des Effets Thérapeutiques, UMR 5558 Université Claude Bernard Lyon1, Lyon, France

**Correspondence to**
Dr Muaamar Al-Gobari; muaamar.algobari@gmail.com

## ABSTRACT

**Objectives** To summarise and synthesise the current evidence regarding the effectiveness of drug interventions to prevent sudden cardiac death (SCD) and all-cause mortality in patients with heart failure with reduced ejection fraction (HFrEF).

**Design** Overview of systematic reviews.

**Data sources** MEDLINE, Embase, ISI Web of Science and Cochrane Library from inception to May 2017; manual search of references of included studies for potentially relevant reviews.

**Eligibility criteria for study selection** We reviewed the effectiveness of drug interventions for SCD and all-cause mortality prevention in patients with HFrEF. We included overviews, systematic reviews and meta-analyses of randomised controlled trials of beta-blockers, angiotensin-converting enzyme inhibitors (ACE-i), angiotensin receptor blockers (ARBs), antialdosterones or mineralocorticoid-receptor antagonists, amiodarone, other antiarrhythmic drugs, combined ARB/neprilysin inhibitors, statins and fish oil supplementation.

**Review methods** Two independent reviewers extracted data and assessed the methodological quality of the reviews and the quality of evidence for the primary studies for each drug intervention, using Assessing the Methodological Quality of Systematic Reviews (AMSTAR) and Grading of Recommendations, Assessment, Development and Evaluation(GRADE), respectively.

**Results** We identified 41 reviews. Beta-blockers, antialdosterones and combined ARB/neprilysin inhibitors appeared effective to prevent SCD and all-cause mortality. ACE-i significantly reduced all-cause mortality but not SCD events. ARBs and statins were ineffective where antiarrhythmic drugs and omega-3 fatty acids had unclear evidence of effectiveness for prevention of SCD and all-cause mortality.

**Conclusions** This comprehensive overview of systematic reviews confirms that beta-blockers, antialdosterone agents and combined ARB/neprilysin inhibitors are effective on SCD prevention but not ACE-i or ARBs. In patients with high risk of SCD, an alternative therapeutic strategy should be explored in future research.

**Systematic review registration** PROSPERO 2017: CRD42017067442.

### Strengths and limitations of this study

► A major strength of our study is that it summarises and synthesises the effectiveness of most evidence-based drug interventions in heart failure patients with reduced ejection fraction for sudden cardiac death (SCD) prevention and classified drug interventions according to the current evidence of their effectiveness.

► Our study used data from published studies and no data from unpublished studies.

► Our study reviews most heart failure drugs on the prevention of SCD and all-cause mortality but limited in scope for not including some drugs such as digoxin, ivabradine and non-drug interventions/devices such as implantable cardioverter defibrillators.

## INTRODUCTION

Heart failure (HF) morbidity and mortality constitute an important burden for patients and for the healthcare systems in both developed and developing countries.[1] Patients with HF are frequently hospitalised and have a high mortality risk because of a poor prognosis or an unexpected death, termed sudden cardiac death (SCD). In people diagnosed with HF, SCD occurs at 6–9 times the rate of the general population. Almost 20% and 80% of patients die within one year and eight years of initial diagnosis, respectively.[1 2] Risk factors of SCD were reported to be similar to cardiovascular diseases. However, the most studied and proven predictor of SCD in patients with HF is left ventricular ejection fraction.[3] Potential drug interventions in patients with heart failure with reduced ejection fraction (HFrEF) include beta-blockers (BBs), angiotensin-converting enzyme inhibitors (ACE-i), angiotensin receptor blockers (ARBs), antialdosterones or mineralocorticoid receptor antagonists, amiodarone, other

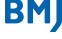

antiarrhythmic agents, combined ARB/neprilysin inhibitors, statins and fish oil supplementation.[4] Some of these interventions aimed at improving survival and reducing total mortality and SCD in HF. For instance, a newly licensed drug (sacubitril/valsartan) in PARADIGM-HF trial (Prospective Comparison of angiotensin neprilysin inhibitor (ARNI) with ACE-i to Determine Impact on Global Morbidity and Mortality in Heart Failure) showed around 20% SCD reduction compared with enalapril.[5]

Nevertheless, optimal strategies for SCD prevention in HF are warranted if we take into account the high portion of mortality that still occurs in this population. Had a practitioner identified a patient with high risk of SCD, it would be important to know which drug is effective or not in SCD prevention other than non-drug interventions such as implantable cardioverter defibrillators (ICDs). However, the large amount of information and the multiple and sometimes discordant systematic reviews on drug interventions could be misleading.[6]

Therefore, it is vital to identify the pharmacological agents that confer the greatest benefit in SCD risk reduction particularly in high-risk patients and if any optimisation of therapeutic strategies to those patients is possible accordingly. Thus, we decided to conduct an overview of systematic reviews to summarise and synthesise the available evidence about the effectiveness of drug interventions in the prevention of SCD in HFrEF and categorised the evidence into effective, ineffective and unclear evidence of effectiveness.

## METHODS

We developed an a priori protocol for this review according to the Preferred Reporting Items for Systematic Reviews and Meta-Analyses statement (online supplementary file S1) and registered it in the PROSPERO International prospective register of systematic reviews (CRD42017067442).

### Data sources and search strategy

Using the Ovid online interface, we searched MEDLINE (up to 24 May 2017), Embase (up to 23 May 2017), ISI Web of Science and the Cochrane Library (up to 24 May 2017). We identified overviews, systematic reviews and meta-analyses of randomised clinical trials by means of a search strategy (available on online supplementary file S2). The search strategy was composed of a filter,[7 8] a mixture of Medical Subject Heading terms (MeSH and EMTREE in MEDLINE and Embase, respectively), text words as well as a truncation when possible without any language or publication date restriction. We did not search conference proceedings nor the grey literature. Reference lists of the included reviews were manually checked for any additional eligible studies. We contacted corresponding reviews' and primary studies' authors to seek for relevant unreported data. If judged necessary, we intended to update the included reviews by searching primary studies published after the systematic review publication date. Apart from authors' expertise in the field, we decided to update if the most up-to-date review of a drug intervention was published more than 5 years ago and/or new clinical trials are not integrated into the evidence.

### Selection criteria and data abstraction

Studies were eligible if they were overviews, systematic reviews and meta-analyses of randomised clinical trials that evaluated the effectiveness of drug interventions in patients with HFrEF. Reviews were included if they examined the effectiveness of the following drugs: BBs, ACE-i, ARBs, antialdosterones or mineralocorticoid receptor antagonists, amiodarone, antiarrhythmics, combined ARB/neprilysin inhibitors, statins and fish oil supplementation. The selected reviews should have contained at least one of the aforementioned HF therapy and had evaluated SCD and/or all-cause mortality prevention as outcomes. We used Endnote and Rayyan[9] to remove duplicates during the selection based on titles and abstracts, and full-text screening.

The abstracted data included eligibility criteria, population type, ejection fraction, study design (including intervention and comparator arms), follow-up duration and authors' evaluation of outcomes. Two reviewers (MA and SA) independently abstracted data. We resolved discrepancies by consensus or by adding a third reviewer's judgement when necessary.

### Quality assessment of the included reviews
#### Methodological quality of the included reviews

Two authors (MA and SA) independently used the AMSTAR (Assessing the Methodological Quality of Systematic Reviews) measurement tool to assess systematic reviews included in our overview. The AMSTAR checklist comprises 11 questions (online supplementary S3 table) and each question accounted for one score point.[10] The answer of 'yes' gave a score of 1 and zero otherwise. This increasingly adopted tool was used at the data collection step as stipulated in the overview protocol.[11] If the authors of included reviews failed to publish their protocol, we deducted a score of one. In addition, we scored 'yes' if the authors mentioned that two reviewers were involved in the study screening, selection or data extraction.

### Quality of evidence in the included reviews

Two authors (MA and SA) independently used the Grading of Recommendations, Assessment, Development and Evaluation (GRADE) approach[12] to assess the quality of evidence of each intervention. GRADE is a widely accepted tool that allows the assessment of five key elements: risk of bias, inconsistency, indirectness, imprecision and publication bias. GRADE categorises the quality of evidence into four levels: high, moderate, low and very low. In the presence of a high risk of bias, the quality of the evidence is downgraded from high to moderate and so on. We also reported the GRADE assessments reported by the authors of the included reviews, or assessed them

otherwise. Moreover, we did not reassess the risk of bias at primary study level if authors of included reviews had sufficiently assessed their quality. In the case of the updated review of ARBs, however, we assessed the quality of newly added randomised clinical trials and integrated it into the evidence synthesis.

## Statistical analysis and data synthesis

We provided a narrative synthesis of the findings of the included reviews and if multiple reviews existed for the same intervention. However, in the case of ARBs, we updated the evidence and meta-analysed the data using random effects and fixed effects model with Mantel-Haenszel methods[13] and reported random effects model to account for heterogeneity. Meanwhile, we evaluated each intervention against our outcomes of interest and synthesised the evidence taking into account heterogeneity and inconsistencies between reviews. As a rule of thumb, $I^2$ (I-square) values of 25%, 50% and 75% correspond to low, moderate and high levels of heterogeneity, respectively.[14]

For the purpose of our overview, we categorised the evidence of the included interventions into three categories: (1) effective interventions; (2) ineffective interventions; and (3) uncertain evidence (conflicting or inconclusive evidence). We used odds ratios (OR) and relative effect or risk ratio (RR) as a summary statistic from the most recent or largest published systematic reviews, and confidence intervals (CIs) of 95% with a significance level determined at two-sided alpha less than 5%.

## Patient and public involvement

Our study did not involve direct contact with patients or the public.

## RESULTS

### Search result

According to our predefined eligibility criteria, our search strategy in electronic databases and manual searches resulted in 41 studies.[6 15–54] Figure 1 shows the search strategy results. At full-text level, we excluded studies that did not assess our outcome of interest (n=129), were narrative reviews (n=4), did

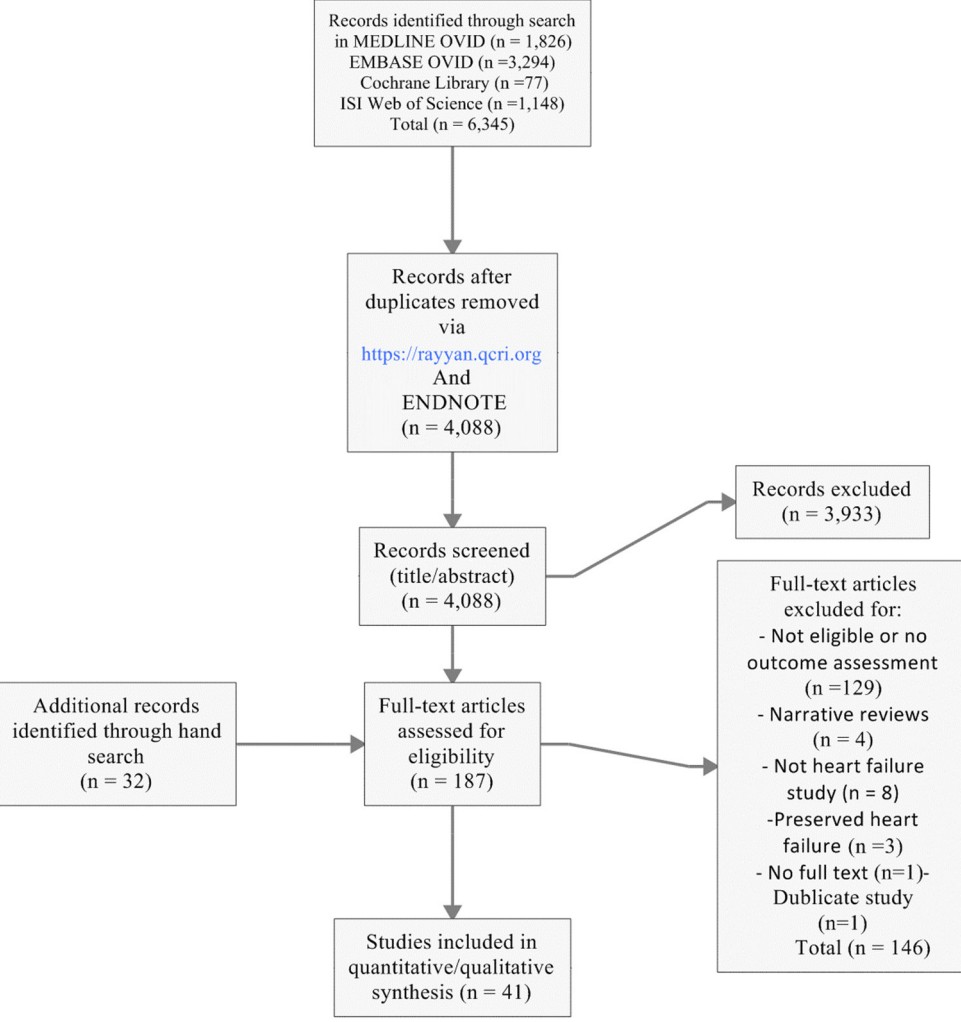

**Figure 1** Flow chart for search result.

not include HF patients (n=8), included preserved patients with HF (n=3), were duplicate or had no full text (n=2).

## Characteristics of the included reviews

As shown in table 1, the population of the included reviews consisted of HF patients with an ejection fraction ≤45% in most studies and a corresponding New York Heart Association classification ranging from I to IV. The effectiveness of each drug intervention has been assessed in at least one review. All reviews were systematic, except two reviews for antiarrhythmic drugs (AADs). At the time of their publication, 15 out of 41 reviews (37%) had corresponding authors based in the USA, 7 (17%) in Canada, 6 (15%) in China, 3 in Chile, 2 in France, 2 in the UK and the 6 remaining in other countries.

The disclosure and reporting of financial resources or funding varied from one study to another. Twenty-one reviews (51%) did not report the source of funding. Ten reviews (24%) reported financial supports that included governments, academic institutions and device industry. Six reviews declared financial resources as none or no external funds. Three reviews reported industry sponsorship for at least one author. One review[53] stated that one author obtained funds for the review without clarifying the source (online supplementary S4 table). We also reported findings summary of each review as stated by their respective authors (table 1).

## Risk of bias and quality of reviews

As shown in table 1, the AMSTAR scores for quality assessment of the included reviews widely ranged from 2 to 10 (out of 11). All reviews had one score less because of non-listing of excluded primary studies except Cochrane reviews,[27 41] which scored 10 because of non-inclusion of grey literature in the search strategy in one review and missing information for funding resources of included primary studies in another, cited respectively (online supplementary S3 table). We did not assess the AMSTAR score for six studies, of which two[46 47] were narrative reviews, two[25 44] were individual participant or patient data meta-analyses and the other two[26 32] were overviews of reviews.

The risk of bias of the included primary studies within reviews remained as judged by the original reviews' authors with the exception of the newly added randomised trials in the update of the ARBs review that we assessed (GRADE) (table 2). The quality of evidence for BBs and antialdosterone agents obtained a high quality on the GRADE scale, while ACE-i, amiodarone and statins obtained a moderate quality. However, combined ARB/neprilysin inhibitors had a moderate and high quality for SCD and all-cause mortality outcomes, respectively, whereas ARBs had a low quality of evidence (table 2).

## Up-to-dateness of included reviews

Most retrieved evidence was published within the last 10 years (2008 and on), and seven (out of nine) drug interventions with updated systematic reviews were within the last 5 years (2012 and on). Moreover, we updated the pooled results for ARBs, which resulted in slightly different results compared with the original Cochrane review.[27]

## Effectiveness of interventions

We report below the summaries of our evaluation on the effectiveness of the drug interventions considered that we have categorised into effective, ineffective and uncertain effectiveness (inconclusive or conflicting evidence).

## EFFECTIVE INTERVENTIONS
### Beta-blockers

Meta-analyses and systematic reviews of randomised controlled trials[15–20] provided overwhelming evidence that BBs decrease the risk of SCD and all-cause mortality in patients with HFrEF. The quality of the evidence was rated high with a relative effect of 0.69 for SCD (OR, 95% CI (0.62 to 0.77)) and of 0.67 for all-cause mortality (OR, 95% CI (0.59 to 0.76)) (table 2).

### Antialdosterone agents

Published studies about mineralocorticoid receptor antagonists (antimineralocorticoids) or (so-called) antialdosterones appeared effective in SCD and all-cause mortality prevention.[21 22 54] However, in a recent systematic review,[21] adverse effects (hyperkalaemia, degradation of renal function and gynaecomastia) were significantly higher in the antialdosterone-treated group compared with placebo. The quality of the evidence was rated high with relative effect for SCD (RR 0.81, 95% CI (0.67 to 0.98)) and all-cause mortality (RR 0.81, 95% CI (0.74 to 0.88)) (table 2).

### ARB/neprilysin inhibitor

One meta-analysis[23] estimated the effects of combined neprilysin renin-aldosterone system inhibition and reported a reduction in SCD and all-cause mortality. The finding was principally derived from one RCT (PARADIGM-HF)[5] that showed about 16% reduction of all-cause mortality in favour of sacubitril/valsartan (LCZ696 previously) compared with enalapril (an ACE-i). This mortality reduction was attributed to a decline on both SCD (20%) and pump failure deaths.[55] Table 2 shows the relative effect for SCD (RR 0.81, 95% CI (0.69 to 0.95)) and all-cause mortality (RR 0.86, 95% CI (0.79 to 0.94)). The moderate quality of the evidence for SCD outcome was due to the estimation from one single clinical trial and the absence of data from other included studies. All-cause mortality was, however, rated as high with a possibility of downgrading

**Table 1** Summary of characteristics of included studies of HF (ordered by intervention)

| Author (year), country | Review type | Intervention/comparator | Population type; ejection fraction (%); NYHA | Study design n; participants n | Mean follow-up/range (months) | Authors' findings summary | AMSTAR score |
|---|---|---|---|---|---|---|---|
| Al-Gobari et al (2013), France[16]* | Systematic review and meta-analysis. | Beta-blockers/placebo; 'usual care'. | HF; <45% except one study <62%; I–IV. | RCTs n=30; n=24779. | Mean: 11.51. | Beta-blockers significantly reduced SCD, cardiovascular death and all-cause mortality. | 6 |
| Chatterjee et al (2013), USA[15] | Systematic review and meta-analysis. | Beta-blockers/placebo; beta-blocker; 'usual care'. | HF; <45%; II–IV. | RCTs n=21; n=23122. | Median: 12. | The study confirmed mortality benefits of BBs compared with placebo or usual care in HF with reduced ejection fraction. | 8 |
| Brophy et al (2001), Canada[17] | Meta-analysis. | Beta-blockers/placebo; 'usual care'. | CHF; <45%; I–IV. | RCTs n=22; n=10135. | Range: 3–23. | This study reported a reduction in mortality and morbidity in CHF. | 4 |
| Lee et al (2001), USA[18] | Systematic review and meta-analysis. | Beta-blockers/placebo. | HF; <30%; II–III. | RCTs n=6; n=9335. | Range : 12–23. | The authors recommended use of beta-blockers in HF with reduced ejection fraction and NYHA II–III. | 4 |
| Bonet et al (2000), USA[19] | Meta-analysis. | Beta-blockers/placebo; 'usual care'. | HF; <45%; NA. | RCTs n=21; n=5849. | Median: 6. | Beta-blockers reduce total mortality by reducing pump failure and SCD events. Vasodilating beta-blockers have perhaps greater effects on overall mortality than non-vasodilating agents. | 4 |
| Heidenreich et al (1997), USA[20] | Meta-analysis. | Beta-blockers/placebo; 'usual care'. | HF; <30%; I–IV | RCTs n=17; n=3039. | Range: 3–24. | Beta-blockers significantly reduced all-cause mortality but showed a trend towards better reduction in non-SCD compared with SCD. | 5 |
| Le et al (2016), France[21]* | Systematic review and meta-analysis. | Anti-aldosterone/placebo; 'usual care'. | HF, post-MI; <40%–>50%; I–IV. | RCTs n=25; n=19333. | Range: 3–39.6. | In HF, antialdosterones or mineralocorticoid receptor blockers reduced SCD (subgroup analysis: 5 RCTs), all-cause mortality (subgroup analysis: 10 RCTs) and cardiovascular, all-cause and cardiovascular hospitalisation. Adverse effects (hyperkalaemia, degradation of renal function and gynaecomastia) were, however, significantly higher in the treated group compared with placebo. | 7 |
| Bapoje et al (2013), USA[54] | Systematic review and meta-analysis. | Antialdosterone/placebo; 'usual care'. | HF; <45%; I–IV. | RCTs n=8; n=11875. | Range: 3–24. | Mineralocorticoid receptor antagonists (or aldosterone antagonists) reduced the risk of SCD in patients with left ventricular dysfunction. | 8 |
| Wei et al (2010), China[22] | Meta-analysis. | Antialdosterone/placebo; 'usual care'. | HF; <45%; NA. | RCTs n=6 (two are not double blind); n=00 000. | Range: 2–24. | Two [67 68] of the six included studies showed a significant reduction of SCD in the group of spironolactone versus placebo and the group of eplerenone versus placebo cited respectively. | 5 |
| Solomon et al (2016), USA[23]* | Meta-analysis. | Sacubitril; valsartan/ACE-i. | HF; <30%; II–IV. | RCTs n=3; n=14742. | Range: 6–27. | The authors concluded that combined neprilysin/RAS inhibition reduced all-cause mortality in HFrEF. | 7 |
| Flather et al (2000), Canada[25] | Systematic review. | ACE-i/placebo. | CHF; post-MI <45%; NA. | RCTs n=5; n=12763. | Range: 15–42. | This meta-analysis showed a lower risk of death in ACE-i treated group compared with placebo. | NA |
| Garg et al (1995), Canada[24]* | Systematic review and meta-analysis. | ACE-i/placebo. | CHF; <45%; I–IV. | RCTs n=32; n=7105. | Range: 3–42. | Overall, this study reported a significant reduction of total mortality (attributed mainly to less progressive HF deaths) and hospitalisation for worsening HF. | 2 |

Continued

**Table 1** Continued

| Author (year), country | Review type | Intervention/comparator | Population type; ejection fraction (%); NYHA | Study design n; participants n | Mean follow-up/range (months) | Authors' findings summary | AMSTAR score |
|---|---|---|---|---|---|---|---|
| Rain and Rada (2015), Chile[26] | Systematic review. | ARB/ACE-i. | HF; <45%–<35%; II–IV. | RCTS=8; n=5201. | NA | The authors concluded that ARBs are probably as effective in mortality reduction as ACE-i with probably less withdrawal rate due to adverse effects. | NA |
| Heran et al (2012), Canada[27]* | Systematic review and meta-analysis (Cochrane). | ARB (or ARB+ACEi) placebo; ACE-i. | HF; ≤40%; ; II–IV. | RCTS n=24; n=25051. | Range: 1–49.5. | Compared with placebo or in addition to ACE-i, ARBs did not reduce all-cause mortality. | 10 |
| Shibata et al (2008), Canada[28] | Systematic review and meta-analysis | ARB/placebo; ACE-i. | HF; <40%; I–IV. | RCTs n=7; n=27495. | Range: 11–41. | Compared with ACE-i or used in combination, ARBs provided no beneficial effects on mortality. A 17% reduction in hospitalisations was observed. | 4 |
| Lee et al (2004), USA[29] | Meta-analysis. | ARB/placebo; ACE-i. | CHF, AMI; ≤45%; II–IV. | RCTs n=24; n=38080. | Range: 1–41. | Compared with ACE-i, ARBs do not differ in efficacy for reducing all-cause mortality in CHF and AMI patients. | 7 |
| Dimopoulos et al (2004), UK[30] | Meta-analysis. | ARB/placebo; ACE-i. | CHF; <40%; II–IV. | RCTs n=4; n=7666. | Mean: 31. | ARBs can be used to prevent events in ACE-i-treated HF patients who are not suitable for beta-blockers. | 3 |
| Jong et al (2002), Canada[31] | Systematic review and meta-analysis. | ARB (or ARB+ACEi)/ placebo ; ACE-i. | HF; ≤35%–≤45%; II–IV. | RCTs n=17; n=12469. | Range: 1–23. | The authors could not conclude any superiority of ARBs versus controls, stating this might be due to the use of ACE-i as a comparator or background treatment in the majority of included trials. | 8 |
| Rain and Rada (2017), Chile[32] | Systematic review. | Statins/placebo; 'usual care'. | HF; <45%; I–IV. | RCTs n=25; n=NR. | NA | The authors summarised that statins do not decrease mortality in chronic HF and might lead to a small reduction in hospital admissions for HF. | NA |
| Al-Gobari et al (2017), Switzerland[6]* | Systematic review and meta-analysis. | Statins/placebo; 'usual care'. | HF, ischaemic/non-ischaemic; NA; I–IV NA. | RCTs n=24; n=111463. | Range: 1–46.8. | Statins do not significantly reduced SCD and all-cause mortality. They may or may not reduce hospitalisations due to worsening HF. | 7 |
| Bonsu et al (2015), Malaysia[33] | Meta-analysis. | Statins/placebo; 'usual care'. | HF; <45%; I–IV. | RCTs n=13; n=10966. | Range: 3–46.8. | Lipophilic statins showed significant decrease in all-cause mortality, cardiovascular mortality and hospitalisation for worsening HF. | 8 |
| Wang et al (2014), China[34] | Meta-analysis. | Statins/placebo; 'usual care'. | HF; NA; NA. | RCTs n=6 (9: observational studies); n=10016. | Range: 12–46.8. | The authors concluded that statins reduce SCD and all-cause mortality in HF. | 5 |
| Liu et al (2014), China[35] | Meta-analysis. | Statins/placebo; 'usual care'. | HF; <45%; I–IV. | RCTs n=13; n=1532. | Range: 3–35.5. | The authors reported significant decrease in all-cause mortality but recommended cautios interpretation and further research. | 7 |
| Rahimi et al (2012), UK[40] | Meta-analysis. | Statins/placebo; 'usual care'. | HF, MI, primary prevention, diabetes, ACS, CHD; NA; NA. | RCTs n=37; n=155020. | | Statins have a modest effect on SCD but no substantial protective effect on ventricular arrhythmic events. | 6 |
| Zhang et al (2011), China[36] | Meta-analysis. | Statins/placebo; 'usual care'. | HF; <45%; I–IV. | RCTs n=13; n=10447. | Range: 2–46.8. | This meta-analysis concluded of no difference between treatment groups but benefits may occur in some specific populations and with a specific statin. | 7 |

Continued

**Table 1** Continued

| Author (year), country | Review type | Intervention/comparator | Population type; ejection fraction (%); NYHA | Study design n; participants n | Mean follow-up/ range (months) | Authors' findings summary | AMSTAR score |
|---|---|---|---|---|---|---|---|
| Xu et al (2010), China[37] | Meta-analysis. | Statins/placebo; 'usual care'. | HF; <45%; I–IV. | RCTs n=7; n=540. | Range: 3–31. | The authors suggested that atorvastatin treatment is effective and reduce all-cause mortality and hospitalisation for worsening HF. | 6 |
| Lipinski (2009), USA[38] | Meta-analysis. | Statins/placebo; 'usual care'. | HF; <45%; I–IV. | RCTs n=10; n=10192. | Range: 3–47. | The authors stated that statins are safe and improve LVEF and decrease hospitalisation for worsening HF. | 7 |
| Levantesi et al (2007), Italy[39] | Meta-analysis. | Statins/placebo; 'usual care'. | Secondary prevention; NA; NA. | RCTs n=10; n=22275. | Range: 6–73.2. | Statins were associated with a significant risk reduction for SCD (in secondary prevention settings). | 3 |
| Claro et al (2015), Chile[41]* | Systematic review and meta-analysis (Cochrane). | Amiodarone/placebo; 'usual care'. | Subanalysis: HF; NA; NA. | RCTs n=11; n=5006. | NA | In HF subpopulation, amiodarone showed a statistically significant reduction for SCD but not for all-cause mortality. Authors judged the quality of the evidence for the whole population (primary prevention) as low to moderate and for secondary prevention population as very low. | 10 |
| Santangeli et al (2012), USA[42] | Systematic review. | Amiodarone/placebo. | Cardiovascular disease; NA; NA. | NA | NA | Amiodarone has less favourable net clinical benefits for prophylaxis of SCD because of adverse effects. | 5 |
| Piccini et al (2009), USA[43] | Meta-analysis. | Amiodarone/placebo; 'usual care'. | HF, AMI; <45%; II–IV. | RCTs n=15; n=8522. | Range: 2–12. | In HF subpopulation, amiodarone showed a statistically significant reduction for SCD but not all-cause mortality. | 7 |
| ATMA Investigators (1997)[44] | Meta-analysis. | Amiodarone/placebo; 'usual care'. | Post-MI and CHF; 31%; NA. | NA | Range: 4.8–25.8. | Amiodarone reduced arrhythmic/sudden death in high-risk patients with recent MI or CHF. All-cause mortality decreased by 13%. | NA |
| Sim et al (1997), USA[45] | Meta-analysis | Amiodarone/placebo; 'usual care'. | Subgroup: HF; <45%; NA. | RCTs n=5; n=4125. | Range: 6–45.6. | Amiodarone reduced all-cause mortality in high SCD risk groups. | 5 |
| Das et al (2010), USA[46] | Narrative review. | Antiarrhythmics/placebo; 'usual care'. | Subgroup: HF; NA; NA. | NA | NA | Class I antiarrhythmic drugs (AADs) increased all-cause mortality and SCD in post-MI patients. Amiodarone (class III AADs) decreased or have neutral effect on SCD. Caution is warranted to outweigh risks of proarrhythmia and other adverse effects. | NA |
| Hilleman et al (2001), USA[47] | Narrative review. | Antiarrhythmics/placebo; 'usual care'. | HF; <45%; NA. | RCTs n=6; n=10440. | Range: 6.5–45. | Beta-blockers (bisoprolol, carvedilol and metoprolol) reduced total mortality and SCD in HF. Class I antiarryhthmics increased mortality and SCD in a post hoc analysis of SPAF-I study. Amiodarone had mixed results, and dofetilide did not reduce mortality or SCD. | NA |
| Rizos et al (2012), Greece[48] | systematic review and meta-analysis. | Omega 3 Fatty acids/ placebo; 'usual care'. | Cardiovascular diseases; NA; NA. | RCTs n=20; n=68680. | NA | Omega-3 polyunsaturated fatty acids supplementation were not associated with a lower risk of all-cause mortality or SCD. | 8 |
| Kotwal et al (2012), Australia[49] | Systematic review and meta-analysis. | Omega 3 Fatty acids/ placebo; 'usual care'. | Cardiovascular diseases, HF admissions; NA; NA. | RCTs n=20; n=62851. | Range: 6–72. | The authors concluded that there is no clear effect on total mortality and sudden death outcomes. | 7 |

Continued

**Table 1** Continued

| Author (year), country | Review type | Intervention/comparator | Population type; ejection fraction (%); NYHA | Study design n; participants n | Mean follow-up/range (months) | Authors' findings summary | AMSTAR score |
|---|---|---|---|---|---|---|---|
| Kwak et al (2012), Korea[50] | Meta-analysis. | Omega 3 fatty acids/placebo; 'usual care'. | Secondary prevention of cardiovascular disease; NA; NA. | RCTs n=14; n=20485. | NA | This meta-analysis concluded of insufficient evidence. | 8 |
| Chen et al (2011), China[51] | Meta-analysis. | Omega 3 fatty acids/placebo; 'usual care'. | Cardiovascular disease; NA; NA. | RCTs n=10; n=33429. | NA | Omega-3 fatty acids did not appear to reduce SCD under guideline-adjusted treatment for CVD secondary prevention. | 7 |
| Marik et al (2009), USA[52] | Systematic review. | Omega t3 dietary supplements/placebo; olive oil; corn oil, sunflower oil; 'usual care'. | Cardiovascular disease; NA; NA. | RCTs n=11; n=39044. | Range: 12–55.2. | Dietary supplementation with omega-3 fatty acids reduced SCD and all-cause mortality. | 4 |
| Wang et al (2006), USA[53] | Systematic review. | n–3 Fatty acids/placebo/ olive oil; corn oil, sunflower oil; 'usual care'. | Primary and secondary prevention; NA; NA. | RCTs n=12; n=32981. | Range: 12–48. | The authors concluded of a significant reduction in all-cause mortality and SCD with n-3 fatty acids from fish or fish oil supplements but not α-linolenic acid. | 6 |

ACE-i, angiotensin-converting enzyme ACE inhibitors; ACS, acute coronary syndrome; AMI, acute myocardial infarction; AMSTAR, assessing the methodological quality of systematic reviews ; ARBs, angiotensin receptor blockers ; BBs, beta-blockers; HF, heart failure; CHD, coronary heart disease; CHF, Chronic heart failure; LVEF, left ventricular ejection fraction; MI, myocardial infarction; NR, not reported; NYHA, New York Heart Association classification ; RAS, renin-angiotensin system; RCTs, randomized clinical trials; SCD, sudden cardiac death; SPAF-I, stroke prevention atrial fibrillation study

in case of a detectable publication bias or unestablished class effect.

## INEFFECTIVE INTERVENTIONS
### ACE inhibitors
Although two systematic reviews,[24 25] with an AMSTAR score of 2/11 and 3/11, respectively, reported a decline in total mortality and less progressive HF deaths, SCD events did not significantly decrease (OR 0.91, 95% CI (0.73 to 1.12)). The quality of the evidence was rated as moderate because of the unclear or high risk of bias in included primary studies (table 2).

### Angiotensin receptor blockers
As shown in figures 2 and 3, we updated a Cochrane review[27] by including more eligible primary studies such as SUPPORT trial.[56] Comparing ARBs with controls resulted in a slightly different effect size estimation. Eventually, we did not combine the different control groups to account for heterogeneity. In stratified analyses, ARBs compared with placebo remained ineffective for all-cause mortality (RR 0.79, 95% CI (0.55 to 1.13)). Similarly, ARBs, compared with ACE-i or in combination versus ACE-i alone, were not superior in all-cause mortality reduction (RR 0.87, 95% CI (0.56 to 1.36); RR 0.99, 95% CI (0.90 to 1.09), respectively) (figure 2). The quality of the evidence is rated as low because of risk of bias, imprecision and inconsistency ($I^2 \approx 78$, p=0.010 for SCD outcome) (table 2). Data were limited for studies reporting SCD, in particular those comparing ARBs versus placebo, or versus ACE-i (figure 2). In addition, the funnel plot for all-cause mortality outcome showed no evidence of publication bias (figure 4). No estimation of publication bias and no funnel plot was drawn for SCD as only five studies reported this outcome.

### Statins
Published systematic reviews and meta-analyses about statins in HF were inconsistent[6 32–40] with a recent tendency towards inefficacy in total mortality and SCD prevention. The quality of the evidence was rated as moderate because of a likelihood of publication bias revealed on the most up-to-date systematic review[6] (table 2).

## UNCLEAR EVIDENCE OF EFFECTIVENESS
The evidence of effectiveness of the drug interventions reported below was considered uncertain due either to conflicting or inconclusive evidence.

### Amiodarone and AADs
Recently published systematic reviews[41 43] for amiodarone showed a significant reduction for SCD but not for all-cause mortality with less favourable net clinical benefits.[42] Other older reviews[44 45] of minor quality (AMSTAR of 3/11 and 5/11 cited respectively) reported a decline of both SCD and all-cause mortality.

**Table 2** Summary of findings and GRADE evaluation for sudden cardiac death (SCD) and all-cause mortality prevention

Drug interventions for SCD and all-cause mortality prevention in heart failure patients

| Outcome | Intervention/comparison | Assumed risk with comparator | Corresponding risk with intervention | Relative effect (95% CI) | Number of participants (no. of studies) | Quality of the evidence (GRADE) | Comments |
|---|---|---|---|---|---|---|---|
| SCD | | | | | | | |
| | Beta-blockers/placebo | 77 per 1000 | 54 per 1000 (49–60) | OR 0.69 (0.62 to 0.77) | 24 779 (26 RCTs) | ⊕⊕⊕⊕ High* | $I^2$=0% (p=0.57) |
| | Antialdosterone inhibitor/ placebo; 'usual care' | 61 per 1000 | 49 per 1000 (41–60) | RR 0.81 (0.67 to 0.98) | 8301 (5 RCTs) | ⊕⊕⊕⊕ High* | $I^2$=8% (p=0.36) |
| | ARB; neprilysin inhibitor/ACE-i | 74 per 1000 | 60 per 1000 (51–70) | RR 0.81 (0.69 to 0.95) | 8399 (1 RCsT) | ⊕⊕⊕⊖ Moderate† | |
| | ACE-i/placebo | 59 per 1000 | 54 per 1000 (43–65) | OR 0.91 (0.73 to 1.11) | 6988 (30 RCTs) | ⊕⊕⊕⊖ Moderate‡ | $I^2$=0% (p=0.94) |
| | ARB (or ARB+ACE i)/Placebo; ACE-i | See comment | See comment | Not estimable | 13 884 (5 RCTs) | ⊕⊕⊖⊖ Low‡§ | $I^2$=78% (p=0.010). Overall, we did not pool the studies because of heterogeneity |
| | Statins/placebo; 'usual care' | 108 per 1000 | 100 per 1000 (76–131) (99 per 1000 (72–131)) | RR 0.92 (0.7 to 1.21) (OR 0.90 (0.64 to 1.24)) | 10 077 (8 RCTs) | ⊕⊕⊕⊖ Moderate¶ | $I^2$=42.6% (p=0.094) |
| | Amiodarone/placebo; 'usual care' | 118 per 1000 | 93 per 1000 (79–110) | RR 0.79 (0.67 to 0.93) | 5006 (11 RCTs) | ⊕⊕⊖⊖ Low¶‡ | |
| | Omega 3 fatty acids/placebo; 'usual care' | 93 per 1000 | 88 per 1000 (77–102) | RR 0.94 (0.82 to 1.09) | 6975 (1 RCT) | ⊕⊕⊕⊖ Moderate† | |
| All-cause mortality | | | | | | | |
| | Beta-blockers/placebo | 178 per 1000 | 127 per 1000 (113–141) | OR 0.67 (0.59 to 0.76) | 24 779 (26 RCTs) | ⊕⊕⊕⊕ High* | $I^2$=40 % (p = 0.02) |
| | Antialdosterone inhibitor / placebo; ' usual care' | 200 per 1000 | 162 per 1000 (148–176) | RR 0.81 (0.74 to 0.88) | 9019 (10 RCTs) | ⊕⊕⊕⊕ High | $I^2$= 0% (p= 0.56) |
| | ARB; neprilysin inhibitor /ACE -i | 183 per 1000 | 158 per 1000 (145–172) | RR 0.86 (0.79 to 0.94) | 14 742 (3 RCTs) | ⊕⊕⊕⊕ High | $I^2$= 0% (p = 0.42) |
| | ACE-i/placebo | 219 per 1000 | 178 per 1000 (158–198) | OR 0.77 (0.67 to 0.88) | 7105 (32 RCTs) | ⊕⊕⊕⊖ Moderate | $I^2$=0% (p= 0.95) |
| | ARB (or ARB+ACE -i)/ placebo; ACE-i. | 183 per 1000 | 177 per 1000 (161–197) | RR 0.97 (0.88 to 1.08) | 19 510 (27 RCTs) | ⊕⊕⊖⊖ Low‡** | $I^2$= 24% (p = 0.14) |
| | Statins/placebo; 'usual care' | 273 per 1000 | 240 per 1000 (205–278) (233 per 1000 (199–273)) | RR 0.88 (0.75 to 1.02) OR 0.81 (0.66 to 1) | 11 024 (13 RCTs) | ⊕⊕⊕⊖ Moderate¶ | $I^2$= 37.7% (p =0.083) |
| | Amiodarone/placebo; 'usual care' | 264 per 1000 | 237 per 1000 (211–266) | RR 0.90 (0.80 to 1.01) | 5006 (11 RCTs) | ⊕⊕⊖⊖ Low¶‡ | |
| | Omega 3 fatty acids/ placebo; 'usual care' | 291 per 1000 | 274 per 1000 (253–294) | RR 0.94 (0.87 to 1.01) | 6975 (1 RCT) | ⊕⊕⊕⊖ Moderate | |

*Although graded high, this might be downgraded into moderate if we strictly consider the risk of bias of primary studies other than an overall estimation.
†Estimation comes from one single clinical trial. No data obtained from other relevant studies for this outcome.
‡The studies reported to generally have a moderate to high risk of bias due to allocation concealment and blinding reporting.
¶Likelihood of publication bias presence with an asymmetric funnel plot.
§Inconsistent results ranged from no effect to insignificant increase of events ($I^2 \approx$ 71%).
**Most studies have small sample and wide CIs including no effect with appreciable harm or benefit.
ACE-i, ACE inhibitors; ARBs, angiotensin receptor blockers; GRADE, Grading of Recommendations, Assessment, Development and Evaluation; $I^2$, between-study variance due to heterogeneity; RR, risk ratio.

The quality of evidence for amiodarone was rated as low because of the unclear or high risk of bias and potential publication bias in primary studies (table 2). No systematic review for AADs of other classes or drugs (other than amiodarone) were retrieved. Nevertheless, two narrative reviews[46 47] reported that class I antiarrhythmics increased SCD and all-cause mortality. These narrative reviews called for caution regarding the mixed results of amiodarone and its adverse effects.

## Omega-3 polyunsaturated fatty acids (PUFAs) and fish oil supplementation

No systematic review was exclusively conducted in patients with HF for this intervention. One primary study,[57] known as GISSI-Prevenzione HF, recruited patients with chronic HF and reported a lower mortality events in the n-3 PUFAs group compared with the placebo group. The authors reported an adjusted HR of 0.91 (95.5% CI 0.833 to 0.998), p=0.041). However,

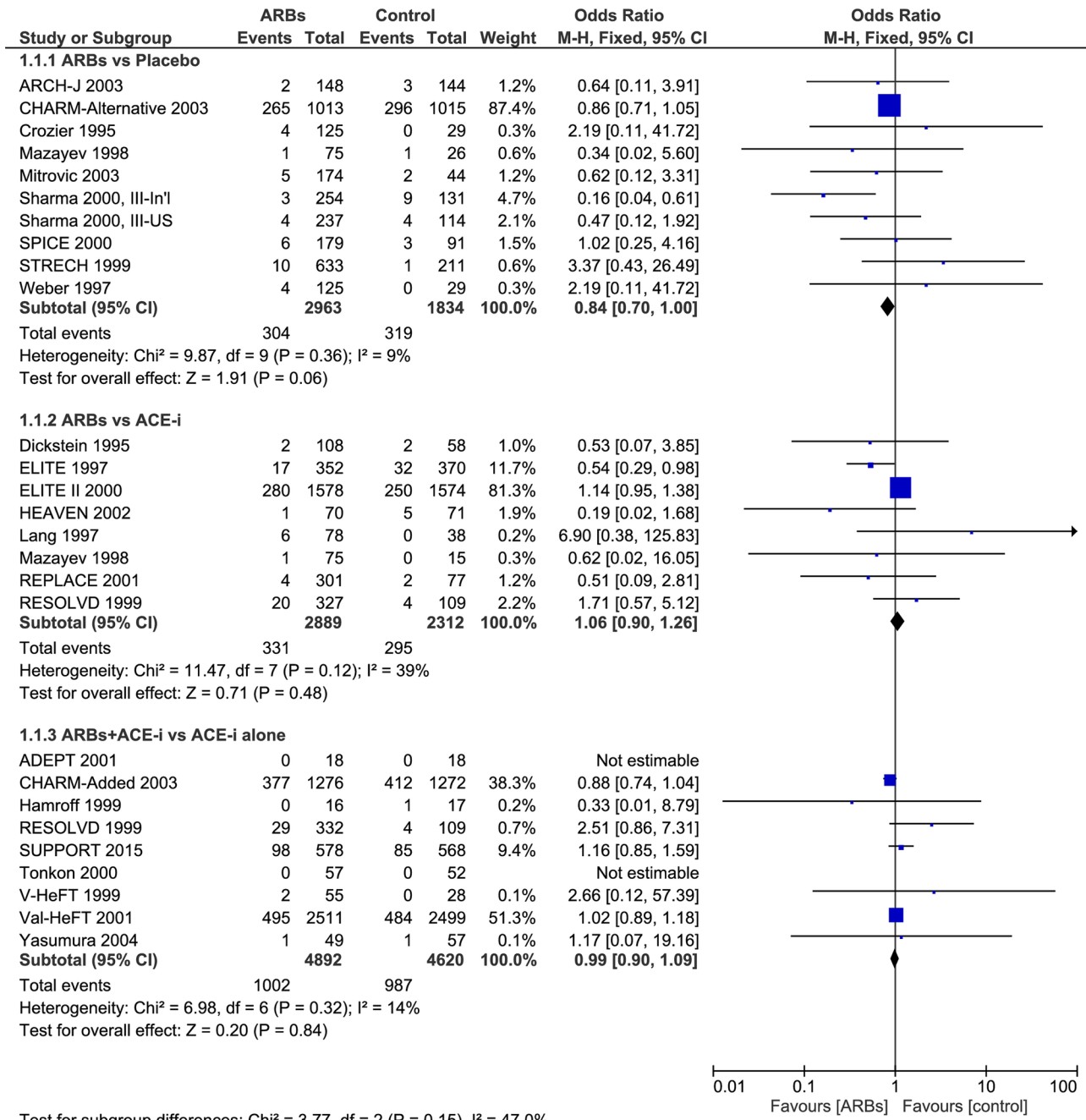

**Figure 2** Efficacy of angiotensin receptor blockers (ARBs) compared with placebo, angiotensin-converting enzyme inhibitor (ACE-i) or combined in heart failure with reduced ejection fraction (HFrEF) for the prevention of all-cause mortality.

relative risk in our analysis remained statistically insignificant (RR 0.94, 95% CI (0.87 to 1.01), p=0.10) and (RR 0.94, 95% CI (0.81 to 1.09), p=0.42) for all-cause mortality and SCD, respectively. Our assessment of the quality of the evidence involving GISSI-Prevenzione HF was moderate because of an absence of data of any other relevant studies (table 2). In addition, some recent systematic reviews[48–51] included patients regardless of their cardiovascular disease and concluded of no clear effect, insufficient evidence or no reduction on SCD and all-cause mortality outcomes. Meanwhile, some older studies[52 53] reported that omega-3 fatty acids

and fish oil supplements (other than α-linolenic acid[53]) reduced SCD and all-cause mortality.

## DISCUSSION

Our assessment of the effectiveness of drug interventions to prevent SCD in patients with HFrEF indicated that BBs, antialdosterone agents, as well as combined ARB/neprilysin inhibitors were effective.

Previously reported meta-analyses and systematic reviews of RCTs[15–20] indicated that BBs are effective in the prevention of SCD and all-cause mortality in HFrEF.

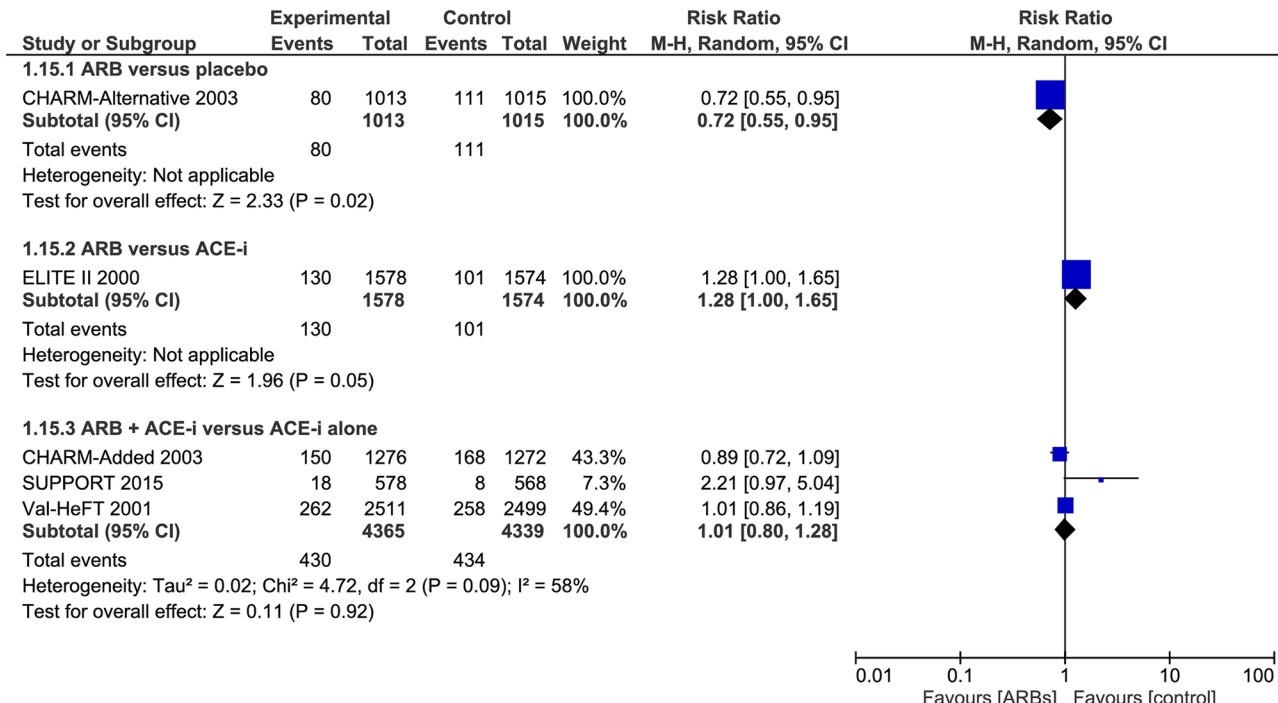

**Figure 3** Efficacy of angiotensin receptor blockers (ARBs) compared with placebo, angiotensin-converting enzyme inhibitor (ACE-i) or combined in heart failure with reduced ejection fraction (HFrEF) for the prevention of sudden cardiac death (SCD).

However, although they were increasingly used as a usual 'routine' care in the compared arms of the more recently published clinical trials,[58] BBs stayed underused for long time and may still be.[59] Mineralocorticoid receptor antagonists or antialdosterone drugs have been reported effective in HFrEF by reducing SCD and all-cause mortality.[21 22 54 60] Our summary of the findings and the consistency of the results support this claim

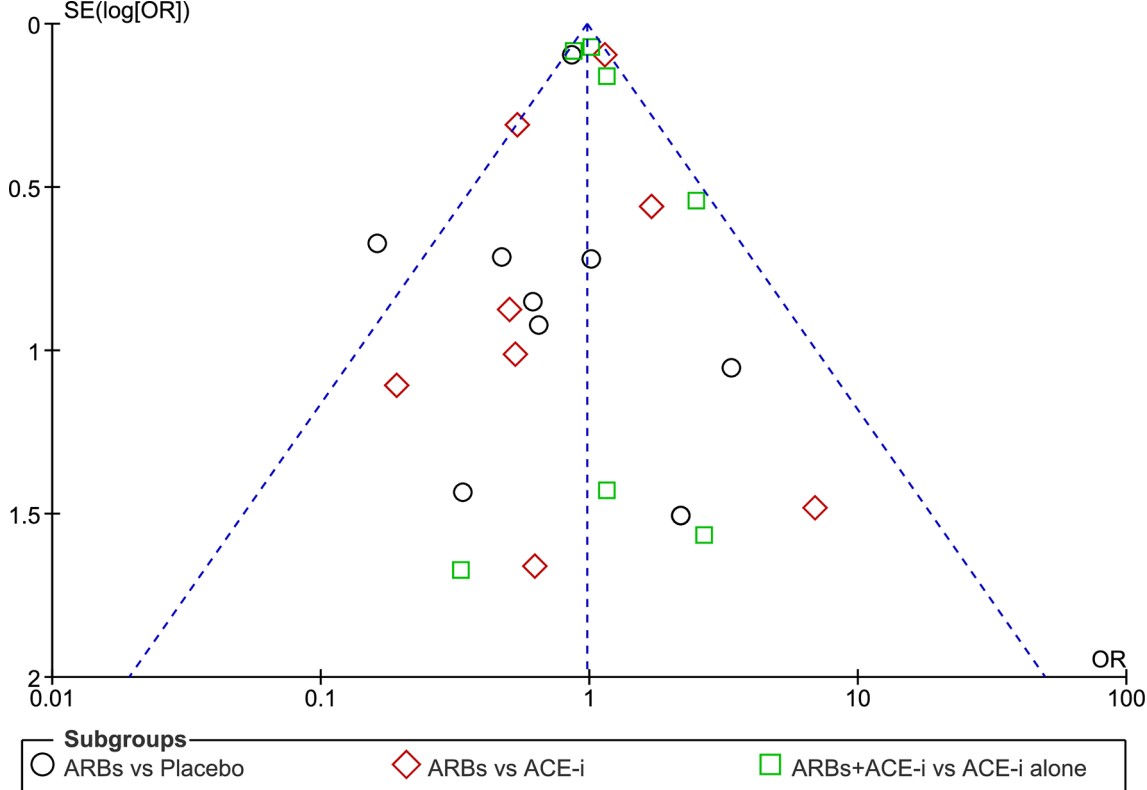

**Figure 4** Funnel plot of SE (log OR) by OR to evaluate publication bias for the efficacy of angiotensin receptor blockers (ARBs) compared with control in heart failure and reduced ejection fraction (HFrEF) for the prevention of all-cause mortality.

with a high quality of evidence. Only one retrieved meta-analysis[23] supported the effectiveness of combined ARB/neprilysin inhibitor. The authors acknowledged the limitation of their meta-analysis, which was not based on a systematic review, but merely pooling three well-known trials published in high impact journals (ie, IMPRESS,[61] OVERTURE[62] and PARADIGM-HF).[5] The quality of the evidence is, however, moderate for SCD and high for all-cause mortality, although our inability to assess any presence of a class effect or a potential publication bias.

We found that ACE-i showed a total mortality reduction in clinical trials and systematic reviews of patients with HF.[24 25] However, our overview showed that ACE-i, surprisingly, did not significantly decrease SCD with a moderate quality of evidence.

In addition, we found that neither ARBs nor statins reduced SCD and/or all-cause mortality. Our findings for ARBs were in agreement with Jong and colleagues,[31] Shibata *et al*[28] and Dimopoulos *et al*,[30] but in contradiction to Lee *et al*[29] and Rain and Rada's conclusions.[26] Our up-to-date meta-analysis for ARBs included only five primary studies, but large-scale trials, that reported SCD events. Eventually, we did not pool all the different comparators together but separately estimated the effect size for each group to account for the heterogeneity. Moreover, the addition of current trials such as SUPPORT[56] improved the statistical power of detecting an effect if existed and the summary statistic remained statistically insignificant (figure 2). Of note, Jong and colleagues[31] attributed this inefficacy of ARBs in HF to the background treatment with ACE-i.

Within the current evidence, ARBs should not be seen as interchangeable with ACE-i, which also showed a neutral effect on SCD, without a proper reason. Therefore, in a high-risk SCD patient, another therapeutic strategy should be sought, and an ARB/neprilysin inhibitor might be an alternative in patients similar to those of the PARADIGM-HF trial.[5]

The addition of statins to the therapy regimen of patients with HF had no survival benefits. Actually, a recent systematic review and meta-analysis indicated that statins did not reduce SCD nor all-cause mortality.[6] Our current study reached the same conclusion with similar quality of evidence.

Our overview showed unclear evidence of effectiveness of omega-3 PUFAs, fish oil supplementation and AADs. The latter intervention had an evidence originated from only narrative reviews, as we did not identify any systematic reviews. Also, only one n-3 PUFA clinical trial[57] was conducted in patients with HF and reported a statistically significant mortality reduction; this result was not supported by other trials and recent systematic reviews,[48–51] a finding that justified our conclusion of unclear evidence. Moreover, no other data or systematic reviews conducted in HF were retrieved by our electronic and manual searches.

AADs are classified into four categories[46]: sodium channel blocking drugs (class I), BBs (class II), potassium channel blockers (class III) and calcium channel blockers (class IV). We found inconclusive evidence of effectiveness of all categories, with the exception of BBs. The evidence of effectiveness of class I, III and IV is inconclusive, neutral or even detrimental to patients as for class I AADs.[46 47] Amiodarones, which present class I, II, III and IV effects, reported mixed results with potential SCD prevention with adverse effects[43] and potentially, but rare,[63] life-threatening proarrhythmias.[46]

Our overview has some limitations. First, we limited the scope of our study to drug treatment, thus excluding devices like ICDs. We believe that non-drug devices should be tackled in future research. European Society of Cardiology (ESC) Guideline (2016)[64] and others (eg, www.uptodate.com) recommend the use of ICDs for only ≤35% of patients with HF and only after optimisation of drug therapy. In fact, SCDs occur in both reduced and preserved HF. Our overview might help to optimise therapy as a first step before introducing ICDs, which applies to a limited HF subpopulation, regardless of costs. Second, we may have failed to include other drug interventions used in HFrEF. Such drug candidates include digoxin, $I_f$-channel blockers (ivabradine), hydralazine/isosorbide dinitrate, nitroglycerin and phosphodiesterase 3 or 5 inhibitors. However, our overview included most commonly prescribed and evidence-based pharmacological therapy in HF as prespecified in our published protocol.[11] Third, we did not use specific drug names in our literature search strategy, in order to avoid omitting a therapy that evaluated SCD and/or all-cause mortality prevention in patients with HF. Fourth, we based our analyses on existing systematic reviews and meta-analyses, and we updated only one meta-analysis. Consequently, we were unable to update the evidence for ACE-i. Furthermore, as indicated by the AMSTAR score, the methodological quality of some of the existing reviews was suboptimal. Fifth, we did not assess the safety of the evaluated drug interventions, nor the contraindications for their prescription, drug–drug interactions, as well as treatment adherence. Indeed, we considered that these important aspects were out of the scope of our analysis. Sixth, we were unable to do a sensitivity analysis, initially suggested in our protocol, for ischaemic versus non-ischaemic HF due to limited data availability. Finally, a potential source of bias relates to authors of this overview being the authors of three of the included reviews.[16 21 65] However, the adopted methodology is in line with systematic reviews guidelines and ensured a double check of data and methodological evaluation by at least two reviewers and a published protocol.[11]

It is noteworthy that high-quality evidence does not necessarily imply strong recommendations, and strong recommendations can arise from low-quality evidence.[66] Therefore, when one intervention is graded high, it is not our intention to say that it is highly recommended, as we did not assess the level of recommendation in our study. In fact, a level of recommendation depends on the strength of evidence and (among others) on values and preferences of

patients, net benefits and cost-effectiveness of a particular intervention.

## Implications for practice

Our study summarises and synthesises the effectiveness of most evidence-based drug interventions in patients with HFrEF for SCD prevention. It classified drug interventions according to the current evidence of their effectiveness. This categorisation could help health professionals and patients making evidence-based decisions based on updated knowledge, particularly whenever a high-risk SCD patient is identified. Currently, there is no an established strategy to deal with patients at high risk of SCD. In such patients, a particular attention should be considered, and a careful selection of available therapeutic options is needed. Furthermore, there might be a shift towards an alternative therapeutic strategy based on SCD prevention-effective drugs in light of our findings.

## CONCLUSION

Our overview indicates that only three drug interventions (BBs, antialdosterones, combined ARB/neprilysin inhibitors) significantly reduce SCD and improve overall survival among individuals with HF and reduced ejection fraction. However, there is no evidence of effectiveness of ARBs to reduce neither all-cause mortality nor SCD (with a low quality of evidence), and ACE-i do not significantly reduce SCD events. When the goal of drug therapy is to reduce SCD, especially in high-risk patients, our synthesis supports the use of the most effective regimen.

**Acknowledgements** We would like to thank Dr Emilie Zuercher for her technical assistance.

**Contributors** Design and conception: MA, FG and BB. MA is the guarantor. Project administration: BB. Writing original draft: MA. Critical analysis: all authors. Data curation: MA and SA. Statistical analysis: MA. Proofread and approved the final draft: all authors.

**Funding** MA, the corresponding author, received a financial grant (2014.0193) from The Federal Department of Economic Affairs, Education and Research (EAER), Switzerland, also from Département universitaire de médecine et santé communautaires (DUMSC), Lausanne university hospital (CHUV), Switzerland.

**Disclaimer** The former funders had no role in study design or conception, study analysis, writing or preparation of the manuscript.

**Competing interests** We declare that some authors of this overview are also authors of some of the included reviews. However, at least two reviewers systematically checked and validated the extracted data including study qualities. We declare that no other competing interests exist.

**Patient consent** Not required.

**Provenance and peer review** Not commissioned; externally peer reviewed.

**Data sharing statement** All data are available in the manuscript and its supported files. Any more information can be requested from the corresponding author.

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
