## [Reviewer comments · BMJ Open]

ARTICLE DETAILS

TITLE (PROVISIONAL)	Effectiveness of drug interventions to prevent sudden cardiac death in patients with heart failure and reduced ejection fraction: an overview of systematic reviews
AUTHORS	Al-Gobari, Muaamar; Al-Aqeel, Sinaa; Gueyffier, François; Burnand, Bernard

VERSION 1 – REVIEW

REVIEWER	Javier Mariani Hospital El Cruce.
REVIEW RETURNED	20-Dec-2017

GENERAL COMMENTS	Introduction Line 33 (page 4): authors state “Moreover, the prescription of poly-drug regimen is usually adopted”. This sentence is confusing since HFrEF patients should be treated with a combination of drugs independently of the SCD risk. Methods Line 53 (page 4): PROSPERO record date is 2 July 2017, and in the “Data sources and search strategy” section the presented search dates are (may 23 and may 24 –both in 2017). Since record date is posterior to search dates is not clear that the protocol was prospective. Please clarify. Lines 52-54 (page 5): “Two authors independently used the AMSTAR measurement tool to assess systematic reviews included in our overview”. The AMSTAR score is not validated for individual data metaanalyses (as are the cases of ATMA and the work by Flather et al). Line 30 (page 6): authors state “...meta-analyzed the data using random-effects and fixed-effects model with Mantel-Haenszel methods”, but figures 2 and 3 (and lines 12 to 30, page 14) only present results for one metaanalytic method (and it’s not specified if is random-effects or fixed-effects model). Line 42-45 (page 6): “We used relative effect [odds ratio (OR) or risk ratio (RR)] as a summary statistic from the most recent or largest published systematic reviews, and confidence intervals (CIs) of 95% with a significance level determined at two-sided alpha less than 5%.” In the case of ACEi, systematic review by Flather et al, is a larger individual data metaanalysis published in 2000 with 12,763 patients and more precise effect estimates than the metaanalysis by Garg et al. (published in 1995) used as “true” effect estimate. Results Table 2: presented estimated effects for beta-blockers are from a metaanalysis with 24,779 patients (26 RCT’s), and for aldosterone antagonists are from a metaanalysis with 9,019 patients (10 RCT’s), but these metaanalyses are not presented in table 1 (in the case of
---

	beta-blockers, there is a metaanalysis with 24,779 patients and 30 RCT's, and this appears to be the study, but is not so clear for aldosterone antagonists). Table 1: There are values of "00000", please correct. Discussion Lines 32-36 (page 15 –first paragraph): the statement "Our assessment of the effectiveness of drug interventions to prevent SCD or all-cause mortality in HFrEF patients indicated that beta-blockers (BBs), antialdosterone agents, as well as combined angiotensin receptor blocker (ARB)/neprilysin inhibitors were effective.", is incorrect, since ACEi reduce total mortality with a moderate quality of evidence (as is the case for ARB/neprilysin inhibitors). Lines 27-29 (page 6): authors state "... in a high risk SCD patient, another therapeutic strategy should be adopted and an ARB/neprilysin inhibitor might be an alternative", could be misleading since such affirmation is based in one trial with strict inclusion criteria (so, this conclusion is applicable to patients who also meet inclusion criteria for PARADIGM-HF trial). Lines 7-11 (page 18): same comment as above for the sentence "When the goal of drug therapy is to reduce SCD, especially in high-risk patients, our synthesis supports the option of replacing ACE-i by combined angiotensin receptor blockers/neprilysin inhibitors. This option has already been proposed by the 2016 ESC guidelines". This statement could be misleading. Abstract Conclusion: same comment as above for the statement "In high-risk SCD patients, a shift from ACE-i to a combined ARB/neprilysin inhibitor is an option". This sentence should be eliminated from abstract.
--	---

REVIEWER	Natale Daniele Brunetti University of Foggia, Foggia, Italy
REVIEW RETURNED	22-Dec-2017

GENERAL COMMENTS	We read with great interest this meta-analysis on Effectiveness of drug interventions to prevent sudden cardiac death in patients with heart failure and reduced ejection fraction. The analysis is well managed and results novel. Please provide Forest plots for beta-blockers and anti-aldosteron agents to further improve the quality of the paper.
---

REVIEWER	Kotaro Nochioka Tohoku University Hospital
REVIEW RETURNED	25-Dec-2017

GENERAL COMMENTS	This paper evaluated the value of medications on prevention of sudden cardiac death (SCD) and all-cause mortality in patients with heart failure and reduced ejection fraction (HFrEF). The authors concluded that that beta-blockers, anti-aldosterone agents and combined ARB/neprilysin inhibitors are effective on SCD prevention but not ACEI or ARBs. First, I appreciate for the authors' meticulous efforts. In this study, the search for relevant studies was exhaustive. The selection and assessments of studies are reproducible.
---

	I have the following points for the authors to address:  1. I think not medications, but ICD is most important for SCD prevention. Using ACEI, beta-blockers and anti-aldosterone agents for HFrEF is the standard therapy in this era. So, I am not sure the clinical significance of this paper. 2. Weakness is the confirmatory nature of the findings in light of prior studies, which limit novelty and impact, even though the authors claim “up-to-date” analysis.
--	--

REVIEWER	Federico Guerra Marche Polytechnic University
REVIEW RETURNED	29-Dec-2017

GENERAL COMMENTS	Muaamar and colleagues try to summarize the effects of drug interventions on prevention of sudden cardiac death (SCD) and all-cause mortality in patients with HFrEF by using an umbrella review. The paper is interesting and provides some clever insights, but the main strength of the present manuscript lies surely in the methodology. The review was performed with a priori protocol following the PRISMA statement and registered in the PROSPERO registry. Study selection and data extraction process are optimally performed and reported. I have only three concerns:  - There's a methodological issue that arises with all umbrella reviews: the older the paper, the more times it gets into a meta-analysis, hence the more weight it will have in umbrella reviews. How did the authors cope with this resonance effect? One solution could be to perform a sensitivity analysis in order to detect a time-dependent shift of the relative effect for each drug considered. - The results of ACE-Is and ARBs on SCD were quite disappointing. However, one potential bias should be discussed: according to table 2, ACE-I were compared with placebo and were older studies, where the absolute risk of SCD was higher because the standard of care was lower. ARBs were compared with either placebo or ACE-I, and the standard of care was higher as the papers were more recent. Therefore, the take-home message could also be that ARBs do not add anything when compared to ACE-I. I would suggest to split the comparisons on SCD and deliver two distinct relative effects (ARBs vs. placebo and ARBs vs. ACE-I), discussing each other separately. Another (much less interesting) option would be just to mention the present limitation and rework the abstract and the discussion accordingly (the resulting I2 also supports this type of approach). - As the authors state, there are no mention of non-pharmacological therapies like ICD or CRT devices, which could however influenced the results. Did any of the meta-analysis included report the prevalence of such devices in their population? It should be interesting to provide these data by adding a column on table 1. Minor comments:  - Table 1: the range of the paper by Lee S. et al is wrong.
---

REVIEWER	Dr Pupalan Iyngkaran Northern Territory Medical School, Flinders University
REVIEW RETURNED	30-Dec-2017

GENERAL COMMENTS	Thank you for submitting an important and timely paper on this topic. IT arrives at the advent of the introduction of ARB/Nepriylisin inhibitor
---

and its mechanism for >20% reduction in MACE over ACE-i and additional benefits needs to be better deciphered. The scope and breadth with the agents covered is vast. All-in-all a high quality presentation of information.

There are some small typo-logical and grammatical errors (e.g pg 5-41)

PS// I suggest none of the data is omitted. The additional data that does not fit into the main article should be part of a supplementary. The paper trail (methodological description) for work to follow could be very useful.

Abstract - contents good. Results - more details including actual values can be considered. Please clarify if this the format and headers preferred by the journal. Check maximum number of key words allowed. With the details provided and attention to detail the article provides a compelling case for publication. Conc: "ACE-I do not reduce SCD",? please check as Table 2 suggests trends, although CI >0.73 o 1.11 there were 30 studies; any difference with newer type of ACEI or ischemic HF studies, higher risks, etc?

Intro: Adequate

- pg 4-26 to 33. This paragraph needs to be reviewed. 1) "SCD treatment is sub-optimal" - needs a reference. It is better worded as 'SCD can be better risk stratified with improved understanding of therapeutics including more novel agents. he advent of devices have changed this, however evidence may also suggest superiority of newer agents.' 2) "large amount of info misleading...." please reference or edit. 3) "poly-drug regimen is adopted,..." please edit - as this is the mainstay of HF treatment. Better worded '..with advent of ARB/NRI developing evidence for changing to this novel class of agent within a class needs further study, as it would among other HF drug classes'

Methods: Adequate

- Please check if the format and headings of the journal on this section are met.

Fig 1 good. Please explain nature of 146 studies excluded. Were there overlap in studies, were there studies excluded, could this exclusion skew your findings. e.g It is surprising to see ACE-I not having an impact on reducing SCD! More likely the effect could be less than ARB/NRI. Need to tease out the 30 RCT's e.g ischemic HF, lower EF, etc.

Results: Adequate

- Search result - please expand. Perhaps more on omissions - how this could skew/bias results (or discuss later?)

- MADIT and device studies show highest risk of SCD with lower drug use, worse EF, NYHA class and ishemic HF. Were all the various drug arms comparable on this on draw conclusions?

- Table 1 & 2 - excellent. Im not clear hot to interpret the comments column. Please clarify better.

- Breakdown into sections by drug classes and descriptions - good.

- Figures 2-4 - are there data for other drugs as well? Why did you only choose RAAS agents? Can you present one plot/graph with all the agents you describe summarizing the total participants, no of trials and statistical significance?

Discussion: good

	Disclosures to References - good
REVIEWER	Jimmy Efir University of Newcastle, Australia
REVIEW RETURNED	28-Feb-2018
GENERAL COMMENTS	This is a nicely written manuscript. My comments are mostly minor. 1) It is important to differentiate between stable and unstable heart failure. 2) Ideally, the analyses should be presented separately according to the relative effects estimates that were used when reporting results. For example, ORs, RRs, HRs, etc. are scaled differently and it is best not to group them as if they are the same measure. 3) Organic nitrate esters such as nitroglycerin continue to be used in the treatment of heart failure. Also, it is important that the authors consider PDE-3 inhibitors such as Amrinone, milrinone, and enoximone, which are used to treat heart failure.

VERSION 1 – AUTHOR RESPONSE

Reviewer: 1

Reviewer Name: Javier Mariani

Institution and Country: Hospital El Cruce.

Please state any competing interests: None declared

Please leave your comments for the authors below

Introduction

Line 33 (page 4): authors state “Moreover, the prescription of poly-drug regimen is usually adopted“. This sentence is confusing since HFREF patients should be treated with a combination of drugs independently of the SCD risk.

We thank the reviewer for the comments. This sentence has been removed.

Methods

Line 53 (page 4): PROSPERO record date is 2 July 2017, and in the “Data sources and search strategy” section the presented search dates are (may 23 and may 24 –both in 2017). Since record date is posterior to search dates is not clear that the protocol was prospective. Please clarify.

At the time of submission to PROSPERO, we mentioned that only preliminary searches started but not completed. Search strategy was designed. However, no formal screening started, nor data extraction, nor analysis....etc. The whole protocol was submitted on mid-June and the actual start date as mentioned in PROSPERO was 23 May 2018. It takes days to design a search strategy, to do preliminary searches, and to write the protocol and it took few weeks before the protocol being registered and appeared online (2 July 2017). All processes are according to PROSPERO guidelines and to the standard methodology. The current version has been updated after the completion of this overview (12 January 2018).

Lines 52-54 (page 5): “Two authors independently used the AMSTAR measurement tool to assess systematic reviews included in our overview“. The AMSTAR score is not validated for individual data metaanalyses (as are the cases of ATMA and the work by Flather et al).

We modified this in the tables and in the text where necessary.

Line 30 (page 6): authors state "...meta-analyzed the data using random-effects and fixed-effects model with Mantel-Haenszel methods", but figures 2 and 3 (and lines 12 to 30, page 14) only present results for one metaanalytic method (and it's not specified if is random-effects or fixed-effects model).

We modified our manuscript stating that we reported random effects model to account for heterogeneity. In Fig 2 and 3: it is mentioned above the forest plot "M-H, Random, 95% CI" : M-H stands for Mantel-Haenszel and "random" for the random-effects model (some authors write it again in the legend but we think that it is unnecessary).

Line 42-45 (page 6): "We used relative effect [odds ratio (OR) or risk ratio (RR)] as a summary statistic from the most recent or largest published systematic reviews, and confidence intervals (CIs) of 95% with a significance level determined at two-sided alpha less than 5%." In the case of ACEi, systematic review by Flather et al, is a larger individual data metaanalysis published in 2000 with 12,763 patients and more precise effect estimates than the metaanalysis by Garg et al. (published in 1995) used as "true" effect estimate.

We thank the reviewer for such precise remarks. It is true that we used the analysis of Garg et al. as the reference effect estimate because the study by Flather et al. included studies (three of them - out of five- as stated in the abstract) after Myocardial infarction. It is known that post-MI patients are another population. Some of them develop heart failure later but they are still another study population. We quote from the study of Flather et al. " From all the available evidence (including a previous overview¹), we have conclusively shown that use of ACE inhibitors after an acute myocardial infarction provides clear benefit to a wide range of patients". Even though the authors tended to extend their study to heart failure patients, the data do not fully support it. Anyway, we acknowledge that the study of Garg et al. is an older study and we would have liked to update the evidence but no further trials were retrieved. This is added to our manuscript for clarity purposes (Page 11, Line 10), including table 1 correction to account for (post-MI).

Results

Table 2: presented estimated effects for beta-blockers are from a metaanalysis with 24,779 patients (26 RCT's), and for aldosterone antagonists are from a metaanalysis with 9,019 patients (10 RCT's), but these metaanalyses are not presented in table 1 (in the case of beta-blockers, there is a metaanalysis with 24,779 patients and 30 RCT's, and this appears to be the study, but is not so clear for aldosterone antagonists).

We double-checked the Table 2 and the original studies. They are both correct as you stated. However, we modified the number of studies from 26 to 30 to avoid confusion. Actually four studies in the original forest plot were "not estimable" due to absence of events in the compared groups. Concerning aldosterone antagonists, Fig 2 of the article Hai-Ha le 2016 et al. contained a subgroup analysis for heart failure and myocardial infarction separately. (5 studies and 10 studies for SCD and all-cause mortality respectively). We specified this point in table 1 "under authors' summary". For clarity, we added a footnote under table 1 for studies from table 1 that were "considered" in table 2. (Symbolized by ≠ after the reference)

Table 1: There are values of "00000", please correct.

We have corrected this mistake.

Discussion

Lines 32-36 (page 15 –first paragraph): the statement “Our assessment of the effectiveness of drug interventions to prevent SCD or all-cause mortality in HFrEF patients indicated that beta-blockers (BBs), antialdosterone agents, as well as combined angiotensin receptor blocker (ARB)/neprilysin inhibitors were effective.”, is incorrect, since ACEi reduce total mortality with a moderate quality of evidence (as is the case for ARB/neprilysin inhibitors).

We modified our manuscript to state only SCD (as it is the main outcome and the main message of the overview). We understand that the preposition “or” make our statement incorrect. We might have another option to modify it to “both SCD and all-cause mortality” which was meant by the statement. We thank the reviewer for this comment.

Lines 27-29 (page 6): authors state “... in a high risk SCD patient, another therapeutic strategy should be adopted and an ARB/neprilysin inhibitor might be an alternative”, could be misleading since such affirmation is based in one trial with strict inclusion criteria (so, this conclusion is applicable to patients who also meet inclusion criteria for PARADIGM-HF trial).

Lines 7-11 (page 18): same comment as above for the sentence “When the goal of drug therapy is to reduce SCD, especially in high-risk patients, our synthesis supports the option of replacing ACE-i by combined angiotensin receptor blockers/neprilysin inhibitors. This option has already been proposed by the 2016 ESC guidelines”. This statement could be misleading.

Abstract

Conclusion: same comment as above for the statement “In high-risk SCD patients, a shift from ACE-i to a combined ARB/neprilysin inhibitor is an option”. This sentence should be eliminated from abstract.

We understand that the evidence for ARB/neprilysin inhibitor is still limited. However, it is classified as an evidence of category B according to ESC guidelines and our study goes in the same direction. We cannot recommend non-use for this limitation but we can also refer the reader to interpret results with caution and in patients similar to Paradigm-HF patients. The superiority of ARB/neprilysin inhibitor over ACE-I for SCD in Paradigm-HF and in our study is clear but our study adds that ACE-i do not reduce SCDs that let us to support the replacement by ARB/neprilysin inhibitor as a plausible solution in a high-risk SCD patient.

We have also deleted that from the abstract and clarified our statements elsewhere as suggested.

Reviewer: 2

Reviewer Name

Natale Daniele Brunetti: Institution and Country
University of Foggia, Foggia, Italy

Please state any competing interests: none declared

Please leave your comments for the authors below

We read with great interest this meta-analysis on Effectiveness of drug interventions to prevent sudden cardiac death in patients with heart failure and reduced ejection fraction. The analysis is well managed and results novel.

Please provide Forest plots for beta-blockers and anti-aldosterone agents to further improve the quality of the paper.

Our overview also provides a narrative review of drug intervention in HF. We guided the reader to systematic reviews and meta-analyses that are published and are updated and no further trials

appeared thereafter. This is the case for beta-blockers and anti-aldosterones. (cf. table 1 and Table 2). We thank the reviewer for the comments.

Reviewer: 3

Reviewer Name: Kotaro Nochioka

Institution and Country: Tohoku University Hospital

Please state any competing interests: None declared.

Please leave your comments for the authors below

This paper evaluated the value of medications on prevention of sudden cardiac death (SCD) and all-cause mortality in patients with heart failure and reduced ejection fraction (HFrEF). The authors concluded that that beta-blockers, anti-aldosterone agents and combined ARB/neprilysin inhibitors are effective on SCD prevention but not ACEI or ARBs. First, I appreciate for the authors' meticulous efforts. In this study, the search for relevant studies was exhaustive. The selection and assessments of studies are reproducible.

We thank the reviewer for the comments.

I have the following points for the authors to address:

1. I think not medications, but ICD is most important for SCD prevention. Using ACEI, beta-blockers and anti-aldosterone agents for HFrEF is the standard therapy in this era. So, I am not sure the clinical significance of this paper.

As we replied to the associate Editor, ICDs are indicated to prevent SCD. However, "with the increasing use of evidence-based medications, rates of sudden death over time may have diminished such that ICDs may not significantly reduce overall mortality when added to appropriate medical therapy in some groups of patients, such as those with nonischemic cardiomyopathy" (Shen L. et al. 2017 see ref. 57). We acknowledge this as a limitation because we believe that an evaluation of ICDs, CRT and other devices are of importance for future research (cf. response to associate editor).

2. Weakness is the confirmatory nature of the findings in light of prior studies, which limit novelty and impact, even though the authors claim "up-to-date" analysis.

It is true that our conclusion that ESC guidelines have already suggested to replace ACE-i by combined ARB/neprilysin inhibitors might limit the impact but, by contrast, that would increase the confidence. However, we have modified our manuscript (abstract, discussion and conclusion sections) to clarify our result impact. Our result that showed the effectiveness of some HF drugs and the ineffectiveness or uncertain evidence of others are novel. We have also included a section "implication for practice". According to our knowledge, the outcome SCD has not been covered in a single document as our study.

Reviewer: 4

Reviewer Name: Federico Guerra

Institution and Country: Marche Polytechnic University
Please state any competing interests: None declared

Please leave your comments for the authors below

Muaamar and colleagues try to summarize the effects of drug interventions on prevention of sudden cardiac death (SCD) and all-cause mortality in patients with HFREF by using an umbrella review. The paper is interesting and provides some clever insights, but the main strength of the present manuscript lies surely in the methodology. The review was performed with a priori protocol following the PRISMA statement and registered in the PROSPERO registry. Study selection and data extraction process are optimally performed and reported.

We thank the reviewer for the comments.

I have only three concerns:

- There's a methodological issue that arises with all umbrella reviews: the older the paper, the more times it gets into a meta-analysis, hence the more weight it will have in umbrella reviews. How did the authors cope with this resonance effect? One solution could be to perform a sensitivity analysis in order to detect a time-dependent shift of the relative effect for each drug considered.

Thank you for this suggestion. It is true that the relative effect could relatively change with time. This means that accumulation of clinical trials, for instance, chronically could tell us if the relative effect is changing within time. This is termed cumulative meta-analysis. In our overview, we referred the interested reader to the current evidence for each studied drug (cf. table 1). For example, from 1981, it was clear that beta-blockers reduced mortality in post-MI patients.

(<https://www.ncbi.nlm.nih.gov/pubmed/1683604>) and
<https://onlinelibrary.wiley.com/doi/pdf/10.1002/clc.4960160302>

- The results of ACE-Is and ARBs on SCD were quite disappointing. However, one potential bias should be discussed: according to table 2, ACE-I were compared with placebo and were older studies, where the absolute risk of SCD was higher because the standard of care was lower. ARBs were compared with either placebo or ACE-I, and the standard of care was higher as the papers were more recent. Therefore, the take-home message could also be that ARBs do not add anything when compared to ACE-I. I would suggest to split the comparisons on SCD and deliver two distinct relative effects (ARBs vs. placebo and ARBs vs. ACE-I), discussing each other separately. Another (much less interesting) option would be just to mention the present limitation and rework the abstract and the discussion accordingly (the resulting I2 also supports this type of approach).

Both ARBs and ACE-i trials are quite old studies. Before the era of beta-blockers, most old trials were compared to placebo and ARBs and ACE-inhibitors are not an exception. In our re-analysis of ARBs for all-cause mortality outcome, ARBs were separately compared with placebo, ACE-i, and both ARBs/ACE-i. Thus, we updated the evidence for angiotensin receptor blockers (ARBs). As originally published as a Cochrane review, we split the comparisons into three: ARBs versus placebo, ARBs versus ACE-i, and ARBs versus both ARBs and ACE-i. All three comparison reported separately and overall for the outcome all-cause mortality (see figure 2, page 27). As requested, we changed Figure 3 (page 28) concerning SCD outcome to follow the same methodology and modified the text and the legend.

- As the authors state, there are no mention of non-pharmacological therapies like ICD or CRT devices, which could however influenced the results. Did any of the meta-analysis included report the prevalence of such devices in their population? It should be interesting to provide these data by adding a column on table 1.

Although ICDs are important to study in SCD prevention, they are indicated to limited patients and after optimization of drug therapy (cf. 2016 guidelines and www.uptodate.com). They will not influence the result because all studies were randomized clinical trials. This means that whatever the percentage of ICDs use, it will be balanced between the compared groups. We checked for example the Paradigm-HF trial, the use of ICDs were reported 14.9% versus 14.7%. Since our study are *not* at a primary study level, it is irrelevant or technically difficult to report the use of ICDs in each trial. Nevertheless, that might be of interest and should be taken into account in an evaluation of non-drug interventions in HF.

Minor comments:

- Table 1: the range of the paper by Lee S. et al is wrong.

Thank you for this comment. We corrected this error.

Reviewer: 5

Reviewer Name: Dr Pupalan Iyngkaran

Institution and Country: Northern Territory Medical School, Flinders University

Please state any competing interests: No

Please leave your comments for the authors below

Thank you for submitting an important and timely paper on this topic. IT arrives at the advent of the introduction of ARB/Nepriylsin inhibitor and its mechanism for >20% reduction in MACE over ACE-i and additional benefits needs to be better deciphered. The scope and breadth with the agents covered is vast. All-in-all a high quality presentation of information.

We thank the reviewer for the comments.

There are some small typo-logical and grammatical errors (e.g pg 5-41)

We reviewed the whole manuscript and paid attention to grammar and spelling mistakes.

PS// I suggest none of the data is omitted. The additional data that does not fit into the main article should be part of a supplementary. The paper trail (methodological description) for work to follow could be very useful.

We have included all data available. We hope that our work is reproducible and helpful for future research.

Abstract - contents good. Results - more details including actual values can be considered. Please clarify if this the format and headers preferred by the journal. Check maximum number of key words allowed.

We rechecked the journal requirements and modified the abstract according to the editor and your requests.

With the details provided and attention to detail the article provides a compelling case for publication.

Thank you for this positive comment.

Conc: "ACE-I do not reduce SCD",? please check as Table 2 suggests trends, although CI >0.73 o 1.11 there were 30 studies; any difference with newer type of ACEI or ischemic HF studies, higher risks, etc?

It is interesting to study the effects of ACE-inhibitors or ARBs according to severity of the HF, ischaemic or non-ischaemic, other subpopulations. This will be feasible if we possess individual data that we unfortunately do not. We mentioned the absence of a subgroup analysis according to ischaemia as a limitation.

Intro: Adequate

- pg 4-26 to 33. This paragraph needs to be reviewed. 1) "SCD treatment is sub-optimal" - needs a reference. It is better worded as 'SCD can be better risk stratified with improved understanding of therapeutics including more novel agents. The advent of devices have changed this, however evidence may also suggest superiority of newer agents.' 2) "large amount of info misleading...." please reference or edit. 3) "poly-drug regimen is adopted,..." please edit - as this is the mainstay of HF treatment. Better worded '...with advent of ARB/NRI developing evidence for changing to this novel class of agent within a class needs further study, as it would among other HF drug classes'

We modified all these phrases (or deleted) and provided references where necessary.

Methods: Adequate

- Please check if the format and headings of the journal on this section are met.

Fig 1 good. Please explain nature of 146 studies excluded. Were there overlap in studies, were there studies excluded, could this exclusion skew your findings. e.g It is surprising to see ACE-I not having an impact on reducing SCD! More likely the effect could be less than ARB/NRI. Need to tease out the 30 RCT's e.g ischemic HF, lower EF, etc.

We explained in the text the reasons of exclusion of the studies. The exclusion of studies were unlikely to influence the result for the studied drugs due to our methodological approach. However, we acknowledge some limitations: the non-inclusion of some drugs in HF such as digoxins and ivabradine or non-inclusion of non-drug interventions (devices). Regarding studying subpopulations of HF (ischaemic versus non-ischaemic..etc.), we would be able to do so if we had individual data of the included studies (cf. limitations, page 18). In addition, Flather et al. conducted a individual data patient (IPD) meta-analysis for ACE-inhibitors but this study contained majorly patients post-myocardial infarction with and without HF that limited the evidence transfer to HF patients. The 146 studies were excluded as follows: 129 were not eligible or had no outcome assessment. 4 studies were narrative reviews. 8 studies were not HF patients (coronary artery syndrome, diabetes). 3 studies were preserved heart failure. 1 study was duplicate and another had no full text. This is added to the manuscript (page 7)

Results: Adequate

- Search result - please expand. Perhaps more on omissions - how this could skew/bias results (or discuss later?)

We wrote more about the exclusion of studies according to our a priori protocol (http://www.crd.york.ac.uk/PROSPERO/display_record.asp?ID=CRD42017067442).

- MADIT and device studies show highest risk of SCD with lower drug use, worse EF, NYHA class and ischemic HF. Were all the various drug arms comparable on this on draw conclusions?

Our overview included systematic reviews of randomized clinical trials. This means that clinical characteristics and background treatment are balanced between compared groups. They are comparable by definition unless this formed part of inclusion and exclusion criteria. For instance, Paradigm-HF trial (<http://www.nejm.org/doi/full/10.1056/NEJMoa1409077>) had ischaemic

cardiomyopathy of 59.1 % in treatment group versus 60.1 in the control group. Likewise for NYHA and left ventricular ejection fraction (severity of HF). We recall that we mentioned this as a limitation in our study (cf. discussion, for instance, page 8. Line 49-50).

- Table 1 & 2 - excellent. Im not clear hot to interpret the comments column. Please clarify better.

We clarified the comments column in the text. It reports the I-square that measures within-study heterogeneity. As a rule of thumb, I-square values of 25%, 50%, and 75% correspond to low, moderate, and high levels of heterogeneity respectively (<https://www.ncbi.nlm.nih.gov/pmc/articles/PMC192859/>).

- Breakdown into sections by drug classes and descriptions - good.

Thank you for the comment.

- Figures 2-4 - are there data for other drugs as well? Why did you only choose RAAS agents? Can you present one plot/graph with all the agents you describe summarizing the total participants, no of trials and statistical significance?

Thank you for this interesting suggestion. This led us to do a network meta-analysis that would gather all interventions and compare them altogether. In the current methodology, it is not feasible. This is very relevant for a future research.

Discussion: good

Disclosures to References – good

Thank you for the comments

Reviewer: 6

Reviewer Name: Jimmy Efird

Institution and Country: University of Newcastle, Australia

Please state any competing interests: None

Please leave your comments for the authors below

This is a nicely written manuscript. My comments are mostly minor.

We thank the reviewer for the comments.

1) It is important to differentiate between stable and unable heart failure.

You may also mean to differentiate between chronic and acute heart failure. Yes, that is very important. Our study aimed to study the effects of HF drug interventions in heart failure with reduced ejection fraction. They are patients who were recruited in clinical trials according to inclusion and exclusion criteria of each trial. This applies to studied drugs: beta-blockers, ARBs, ACE-inhibitors, MRAs,..etc. We considered that in our study analysis

2) Ideally, the analyses should be presented separately according to the relative effects estimates that were used when reporting results.

For example, ORs, RRs, HRs, etc. are scaled differently and it is best not to group them as if they are the same measure.

Our overview included systematic reviews and meta-analysis of the HF drugs. We reported exactly the same effect size measure we found in retrieved studies. Effect size measures are commonly used interchangeably. Although risk ratios, for instance, are more easily interpreted than odds ratios, they are *not* necessarily interpreted the same (<https://www.theanalysisfactor.com/the-difference-between-relative-risk-and-odds-ratios/>) <https://www.stat-d.si/mz/mz13.1/p4.pdf> except if we have rare events, they are almost identical. Generally, this is the case in our study.

3) Organic nitrate esters such as nitroglycerin continue to be used in the treatment of heart failure. Also, it is important that the authors consider PDE-3 inhibitors such as Amrinone, milrinone, and enoximone, which are used to treat heart failure.

Thank you for this comment. We acknowledged that we have missed some drug interventions in our study. This is now written in the limitation part of the discussion (page 18).

VERSION 2 – REVIEW

REVIEWER	Kotaro Nochioka Tohoku University Hospital, Japan
REVIEW RETURNED	16-Apr-2018
GENERAL COMMENTS	The authors addressed adequately. I do not have further comments.
REVIEWER	Javier Mariani Hospital El Cruce, Florencio Varela, Buenos Aires, Argentina.
REVIEW RETURNED	17-Apr-2018
GENERAL COMMENTS	Authors appropriately modified the paper according to commentaries.
REVIEWER	Natale Daniele Brunetti University of Foggia, Italy
REVIEW RETURNED	22-Apr-2018
GENERAL COMMENTS	no further comments on.
REVIEWER	Jimmy Efird University of Newcastle, Australia
REVIEW RETURNED	02-May-2018
GENERAL COMMENTS	My concerns have been addressed in the revised manuscript.
REVIEWER	Pupalan lyngkaran Flinders University, Australia
REVIEW RETURNED	22-May-2018
GENERAL COMMENTS	Overall: Review of sufficient quality to be accepted. Several major areas need review. 1) Editorial team to review sentence construction and phrases and provide English editing support. 2) Clinical Cardiologist/Physician from authors institute to review validity of some general clinical statements made. 3) Title: Minor sentence phrase could be improved. Substitute 'to' for 'in'? Effectiveness of drug interventions " to" prevent sudden cardiac

	death in patients with heart failure and reduced ejection fraction: an overview of systematic reviews Abstract: Minor grammatical. Generally OK. Remove comment "In patients with.....alternative therapy ...explored..." AICD are the gold standard, it is unlikely cardiology community will invest in novel pharmacological agents for SCD. Introduction: - Paragraph 2 edit to ensure accuracy with current cardiology guidelines. All patients will get BB and RAAS from guidelines. No agents are selected purely on SCD basis. There are also clear guidelines on utility of AICD. - Paragraph 3 first line "Therefore there is a need to" This line is not accurate. It could be worded as " It is vital to know the pharmacological agents that confer greatest benefit in SCD risk reduction. Knowing this information could greatly enhance the future direction of device SCD/guidelines or enhance risk stratification in areas where AICD availability is limited." Provide short description of risk factors for SCD (LVEF, NYHA etc) and guidelines for AICD to reduce SCD. Expand on this in discussion and results. Methods: OK. Statistical team to also review. Results/Discussion: -Flow Chart of search strategy is needed to accompany written description. - Provide greater description on range of NYHA; EF; etiology of CCF (IDCM v NiDCM) and how it correlates with risk of SCD. Were the population in each drug class of the same risk? how did they differ? Could this alter finding? - the correlation between ACM and SCD. Is SCD relevant if ACM is improved vs SCD improved but ACM not. - In short 1: are we comparing apples with apples; if not what standards will we use to make the comparison. Only then can we make a satisfactory conclusion on efficacy between agents. Conclusion: explore after modifications made
--	--

VERSION 2 – AUTHOR RESPONSE

Reviewer(s)' Comments to Author:

Reviewer: 3

Reviewer Name: Kotaro Nochioka

Institution and Country: Tohoku University Hospital, Japan

Please state any competing interests: None declared.

Please leave your comments for the authors below

The authors addressed adequately. I do not have further comments.

We thank the reviewer for the comments.

Reviewer: 1

Reviewer Name: Javier Mariani

Institution and Country: Hospital El Cruce, Florencio Varela, Buenos Aires, Argentina.

Please state any competing interests: None.

Please leave your comments for the authors below

Authors appropriately modified the paper according to commentaries.

We thank the reviewer for the comments.

Reviewer: 2

Reviewer Name: Natale Daniele Brunetti

Institution and Country: University of Foggia, Italy

Please state any competing interests: none to disclose

Please leave your comments for the authors below

no further comments on.

We thank the reviewer for the comments.

Reviewer: 6

Reviewer Name: Jimmy Efid

Institution and Country: University of Newcastle, Australia

Please state any competing interests: N/A

Please leave your comments for the authors below

My concerns have been addressed in the revised manuscript.

We thank the reviewer for the comments.

Reviewer: 5

Reviewer Name: Pupalan Iyngkaran

Institution and Country: Flinders University, Australia

Please state any competing interests: None

Please leave your comments for the authors below

Overall: Review of sufficient quality to be accepted. Several major areas need review. 1) Editorial team to review sentence construction and phrases and provide English editing support.

We thank the reviewer for the comments. We made sure that the manuscript is written using an acceptable English language.

2) Clinical Cardiologist/Physician from authors institute to review validity of some general clinical statements made.

We rephrased our statements whenever necessary to make sure that the interpretation of findings are in conformity with the available data. It may be necessary to indicate here that our overview is in line with Cochrane standard methodology. Our authors' list included a consultant cardiologist (FG).

3) Title: Minor sentence phrase could be improved. Substitute 'to' for 'in'?

Effectiveness of drug interventions " to" prevent sudden cardiac death in patients with heart failure and reduced ejection fraction: an overview of systematic reviews

We kindly disagree with the previous comment. After thorough discussion, we decided to keep the title unchanged.

Abstract: Minor grammatical. Generally OK.

Thank you for the comment. We reviewed the abstract grammatically.

Remove comment "In patients with.....alternative therapy ...explored..." AICD are the gold standard, it is unlikely cardiology community will invest in novel pharmacological agents for SCD.

We kindly disagree with the previous comment. SCD is a fatal outcome and represents almost 50% of deaths in heart failure with reduced ejection fraction (see Ref 2: <http://circres.ahajournals.org/content/95/8/754.long>). Further research and development/investment is always welcome and warranted. At least if we want to further reduce the high mortality in this vulnerable population, we should go in this direction.

Introduction:

- Paragraph 2 edit to ensure accuracy with current cardiology guidelines. All patients will get BB and RAAS from guidelines. No agents are selected purely on SCD basis. There are also clear guidelines on utility of AICD.

We partly agree on this comment. However, we believe that patients, who might be identified as having a high risk of SCD, should be treated with a special attention to such fatal outcome and taking into consideration patient's preference, costs and other factors on

the decision-making process. Our study did not compare drug interventions to ICDs. As stated in our response to reviewer 3 and the associate editor on the previous review round, ICDs and other devices or non-drug interventions (such as CRT) were not part of our study at protocol stage and are mentioned as a limitation. It might be noteworthy to recall that ICDs are only recommended after at least 3 month of optimal treatment. Our study aims to provide an overall synthesis on the effectiveness of drug interventions to prevent SCD.

- Paragraph 3 first line "Therefore there is a need to" This line is not accurate. It could be worded as " It is vital to know the pharmacological agents that confer greatest benefit in SCD risk reduction. Knowing this information could greatly enhance the future direction of device SCD/guidelines or enhance risk stratification in areas where AICD availability is limited."

We rephrased the paragraph to avoid any misunderstanding. We took your suggestion into account. We thank you for the comment.

Provide short description of risk factors for SCD (LVEF, NYHA etc) and guidelines for AICD to reduce SCD. Expand on this in discussion and results.

We wrote a short sentence about risk factors of SCD. Our study excluded ICDs and mention that in the introduction, discussion and limitations.

Methods: OK. Statistical team to also review.

Thank you for your suggestion. We thoroughly reviewed our manuscript and made sure that it is reproducible and made the data available to readers.

Results/Discussion:

-Flow Chart of search strategy is needed to accompany written description.

Our manuscript included a flow chart (cf. Fig 1) and a search strategy (cf. S2 file).

- Provide greater description on range of NYHA; EF; etiology of CCF (IDCM v NiDCM) and how it correlates with risk of SCD. Were the population in each drug class of the same risk? how did they differ? Could this alter finding?

Our study is an overview of systematic reviews. All systematic reviews (unless indicated otherwise) included randomized clinical trials. If the etiology of HF is not an exclusion criteria, the study should have similar distribution of patients in the compared groups. If we take the recent study Paradigm-HF (see table 1 in ref. 4), for instance, we find that ischaemic cardiomyopathy is 59.9% in the treatment group versus 60.1% in the control group. NYHA class and EF (or LVEF) were also equally distributed among compared groups. Nevertheless, it would be interesting to calculate the control risk of the different classes if one wants to compare them.

- the correlation between ACM and SCD. Is SCD relevant if ACM is improved vs SCD improved but ACM not.

We thank the reviewer for the comment. We believe that this is an important research question. Our study does not include individual patient data to be able to reply or study this question (see <https://www.ncbi.nlm.nih.gov/pmc/articles/PMC4260412/>).

- In short 1: are we comparing apples with apples; if not what standards will we use to make the comparison. Only then can we make a satisfactory conclusion on efficacy between agents.

In RCTs, patients are randomly assigned to treatment and so are comparable by definition. Therefore, differences in outcomes can be attributed to the intervention/treatment. We evaluated and included only systematic reviews of RCTs. In addition, we did not intend to compare different drug classes among themselves. This might be tackled in future research (using a network meta-analysis). In our approach, we provided a narrative review of systematic reviews that addressed the mortality and SCD reduction in heart failure patients using HF currently recommended drugs.

Conclusion: explore after modifications made

We rephrased the conclusion according to previous comments.